



# Changes in China's anthropogenic emissions during the COVID-19 pandemic

Bo Zheng[1], Qiang Zhang[2,*], Guannan Geng[3], Qinren Shi[3], Yu Lei[4], Kebin He[3]

[1]Institute of Environment and Ecology, Tsinghua Shenzhen International Graduate School, Tsinghua University, Shenzhen
518055, China
[2]Ministry of Education Key Laboratory for Earth System Modeling, Department of Earth System Science, Tsinghua
University, Beijing 100084, China
[3]State Key Joint Laboratory of Environment Simulation and Pollution Control, School of Environment, Tsinghua University,
Beijing 100084, China
[4]Chinese Academy of Environmental Planning, Beijing 100012, China

*Correspondence to*: Qiang Zhang (qiangzhang@tsinghua.edu.cn)

**Abstract.** The COVID-19 pandemic lockdowns led to a sharp drop in socio-economic activities in China in 2020, including reductions in fossil fuel use, industry productions, and traffic volumes. The short-term impacts of lockdowns on China's air quality have been measured and reported, however, the changes in anthropogenic emissions have not yet been assessed
quantitatively, which hinders our understanding of the causes of the air quality changes during COVID-19. Here, for the first time, we report the anthropogenic air pollutant emissions from mainland China during the first eight months of 2020 by using a bottom-up approach based on the near real-time data. The COVID-19 lockdown was estimated to have reduced China's anthropogenic emissions substantially between January and March in 2020, with the largest reductions in February. Emissions of $SO_2$, $NO_x$, CO, NMVOCs, and primary $PM_{2.5}$ were estimated to have decreased by 29%, 31%, 27%, 26%, and
21%, respectively, in February 2020 compared to the same month in 2019. The reductions in anthropogenic emissions were dominated by the industry sector for $SO_2$ and $PM_{2.5}$ and were contributed approximately equally by the industry and transportation sectors for $NO_x$, CO, and NMVOCs. With the spread of coronavirus controlled, China's anthropogenic emissions rebounded in April and since then returned to the comparable levels of 2019 in August 2020. The provinces in China have presented nearly synchronous decline and rebound in anthropogenic emissions, while Hubei and the provinces
surrounding Beijing recovered slower due to the extension of lockdown measures. The reduction ratios of anthropogenic emissions from 2019 to 2020 can be accessed from https://doi.org/10.6084/m9.figshare.c.5214920.v1 (Zheng et al., 2020) by species, month, sector, and province.

## 1 Introduction

The world witnessed the outbreak and spread of the coronavirus disease COVID-19 in the first half of 2020. The widespread
lockdowns to contain the coronavirus include the broad restrictions on travel, business operations, and people-to-people interactions, which have caused an unprecedented disruption in the atmospheric environment. The reduced socio-economic



activities caused an immediate sharp drop in global fossil fuel demand, reduced air pollutant emissions, and cleaned the air (Bauwens et al., 2020; Liu et al., 2020a; Venter et al., 2020). However, unlike the air quality index that is monitored in real-time, the conventional datasets of energy use and air pollutant emissions are only available after one or two years of latency,
which hampers our understanding of the energy-emission-air quality cascade in such a fast-evolving event of COVID-19.

Recently pioneer studies started to explore the new concept of near real-time emission tracking to assess the influence of COVID-19 lockdowns on climate and air quality. These new approaches extrapolated the emission inventories of a baseline year to the current time in 2020 based on observational constraints or relevant activity indicators. The observation-based method ("top-down method") employed air pollutant concentrations measured by satellites (Chevallier et al., 2020; Ding et
al., 2020; Miyazaki et al., 2020; Zhang et al., 2020; Zhang et al., 2021; Zheng et al., 2020) and ground stations (Feng et al., 2020; Xing et al., 2020) to infer the evolution of surface emissions, which are constrained by both observational data and chemical transport models. Satellite imagery of nitrogen dioxide ($NO_2$) is widely used to constrain nitrogen oxide ($NO_x$) emissions due to its broad spatial coverage and high retrieval accuracy. The activity indicator-based method ("bottom-up method") relied on daily electricity generation (Guevara et al., 2020; Liu et al., 2020d; Liu et al., 2020e), confinement index
(Le Quéré et al., 2020), and mobility index (Forster et al., 2020) to estimate the emission changes based on the assumptions associating those activity indicator changes with the anthropogenic emissions. Since few near real-time proxies are available at present, several common datasets have to be used to approximate the emission changes of different source sectors.

The research on near real-time emission tracking is still in its infancy. Substantial gaps exist between what we need to understand the emission dynamics and what the current top-down and bottom-up methods can provide us. The top-down
approach can constrain emission distributions based on real-time observation data, while it lacks sectoral emission details and cannot retrieve all of the reactive species. The bottom-up approach estimates emissions by sector and by species, but it is limited by the lack of the recent emission baseline and sufficient activity data reflecting emissions change. Since each method has its advantages and disadvantages, having both top-down and bottom-up approaches is important at present.

Here, as the second paper following our previous study (Zheng et al., 2020) that estimates China's daily $NO_x$ and carbon
dioxide ($CO_2$) emissions during COVID-19 with a top-down approach, we develop a bottom-up method in parallel to track monthly emissions of all of the conventional air pollutants in mainland China and for the first time report China's anthropogenic emissions from January to August in 2020. We use the Multi-resolution Emission Inventory for China (MEIC) model (Zheng et al., 2018a) to estimate China's emissions in 2018 and 2019 and then use 39 types of near real-time activity data to update the emission estimates to 2020. Provincial and sectoral emissions are estimated by month and the relative
changes in monthly emissions from 2019 to 2020 are compared with the satellite and ground-based observations for a preliminary evaluation. The emission datasets developed in this study can provide the most up-to-date China's emissions input to chemical transport models and help interpret the abrupt changes in pollutant concentrations during the COVID-19 lockdowns.



## 2 Methods and data

### 2.1 Baseline emissions in 2019

We use the MEIC model to estimate China's anthropogenic emissions in 2018 and 2019 following our previous study (Zheng et al., 2018a) that calculated China's 2010–2017 emissions. MEIC is a bottom-up emission model that used the technology-based approach to estimate emissions with activity data, emission factors, and pollution control techniques of emission sources. More than 700 anthropogenic emission sources are included in the MEIC model, which can be aggregated

into five sectors of power, industry, residential, transportation, and agriculture. Power plants (Liu et al., 2015) and cement plants (Liu et al., 2020b) are both treated as point sources in MEIC with detailed facility-level emission parameters and geographical coordinates to estimate emissions and locate their positions. The other industrial plants are estimated as area sources in each province, where the parameters such as emission factors and pollution removal efficiencies are well-tuned using a comprehensive industrial database that includes about 10,000 industrial plants in China (Qi et al., 2017; Zheng et al.,

2017; Zheng et al., 2018b). Emissions from the residential sector are estimated on the base of the survey-based fuel consumption data (Peng et al., 2019), which corrects the underestimation bias of rural coal use statistics in China. Onroad transport emissions are estimated using the county-level information that depicts high-resolution emission distribution patterns at fine spatial scales (Zheng et al., 2014). Off-road transport and agricultural emissions are both estimated as area sources at the provincial level with the detailed provincial activity database and emission factors mainly derived from the

local measurement (Li et al., 2017). Please refer to our previous papers cited above for more details of the MEIC emission model.

### 2.2 Monthly emissions in 2020

We face challenges in estimating monthly emissions of 2020 using the traditional bottom-up method due to a lack of timely updated activity data and emission factors to drive the MEIC model to do a complete calculation. Currently, it is difficult to

achieve real-time information, and both the coal consumption and pollution control statistics are not available until at least one year later. Adapting to such a situation, we develop a new method to update China's monthly emissions from 2019 to 2020 based on the near real-time activity indicators and the emission factor trends of each province. China's emissions of different air pollutants in 2020 are then estimated by source, by month, and by province using the following formula.

$$E_{i,s,p,m_{2020}} = E_{i,s,p,m_{2019}} \times \alpha_{i,s,p,m}$$

$$= E_{i,s,p,m_{2019}} \times \frac{A_{i,p,m_{2020}} \times EF_{i,s,p,m_{2020}}}{A_{i,p,m_{2019}} \times EF_{i,s,p,m_{2019}}}$$

$$\approx E_{i,s,p,m_{2019}} \times \frac{A_{i,p,m_{2020}}}{A_{i,p,m_{2019}}} \times \frac{EF_{i,s,p,2019}}{EF_{i,s,p,2018}}$$

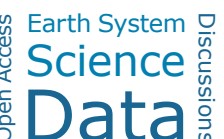

where $i$ represents source, $s$ represents pollutant species, $p$ represents province, $m$ represents month, $E$ is emission, $\alpha$ is the ratio of monthly emissions in 2020 to those in the same month of 2019, $A$ is activity data (e.g., fuel consumption, industrial products), $EF$ is the source-specific net emission factor accounting for pollution removal by end-of-pipe control devices. The pollutant species estimated in this study include sulfur dioxide ($SO_2$), $NO_x$, carbon monoxide (CO), non-methane volatile organic compounds (NMVOCs), and particulate matter whose aerodynamic diameter is smaller than 2.5 micrometers ($PM_{2.5}$),

and black carbon (BC). This equation extrapolates the monthly emissions in 2019 ($E_{i,s,p,m2019}$) to the same month in 2020 ($E_{i,s,p,m2020}$) via the parameter $\alpha$, which is calculated by multiplying the ratio of activity data ($A$) between the two years by the ratio of emission factors. The emission factor ratios between 2019 and 2018 estimated by MEIC in this study are used to approximate the ratios of monthly emission factors between 2020 and 2019 that are not available at present, assuming that each province follows its pollution control pathway that was initiated in 2013 (Zheng et al., 2018a; Zhang et al., 2019).

Monthly emissions in 2019 and emission factor ratios between 2019 and 2018 are derived from the MEIC emission model that has been updated in this study (Sect. 2.1). The activity data ratios are collected and estimated by emission source (Table S1). We collect 39 types of official monthly statistical data (Table S1), of which 22 types are provincial data, to represent the activity changes in different emission sources from 2019 to 2020. The power sector uses monthly thermal power generation. The industrial sector relies on the monthly productions of different products (e.g., iron, steel, cement, glass, and ethylene)

and uses Gross Domestic Product from industrial production to represent the industrial sources without timely updated activity statistics (e.g., industrial fuel burned, industrial solvent use). The residential sector includes commercial, heating, and cooking sources, which use the index of services production, population-weighted heating degree day (Crippa et al., 2020), and the assumption of no change, respectively, to predict the 2019-to-2020 change in the activities. The transportation sector uses the transport volume of freight (tonne • kilometer) and passenger (people • kilometer) by road, railways, and

inland waterways to predict the changes in the oil consumption of vehicles and off-road equipment. The changes in the activity of construction equipment are estimated based on the floor space of real estate that was newly built each month. Besides, we use China's monthly natural gas consumption to constrain the total natural gas use by all of the emission sources.

**2.3 Observation dataset**

We use the observations of surface air pollutant concentrations and satellite $NO_2$ column retrievals to evaluate the estimated

monthly emission changes from 2019 to 2020. The hourly concentrations of air pollutants are derived from the surface measurement network maintained by the China National Environmental Monitoring Center (CNEMC) (http://106.37.208.233:20035/, accessed on 10 November 2020). More than 1500 surface stations are included in the CNEMC observational network at present that covers all of the cities in mainland China. We calculate monthly average concentrations of $SO_2$, $NO_2$, CO, and $PM_{2.5}$ and use the relative changes in these pollutants of the same month from 2019 to

2020 to evaluate the monthly changes in $SO_2$, $NO_x$, CO, and primary $PM_{2.5}$ emissions between these two years. We also evaluate surface $NO_x$ emissions using the $NO_2$ vertical columns, which are retrieved from the TROPOspheric Monitoring



Instrument (TROPOMI) onboard the Copernicus Sentinel-5 Precursor satellite (Veefkind et al., 2012). The official TM5-MP-DOMINO version 1.2/1.3 offline product is used in this analysis (http://www.temis.nl/airpollution/no2col/data/tropomi/, accessed on 10 November 2020).

## 3 Results and discussion

### 3.1 Changes in monthly emissions from 2019 to 2020

China's emissions in 2019 reveal an evident seasonal variation with emissions declining from January to August and a low emission value in February (Table 1 and Figure 1). The emission seasonality is shaped by more emissions from residential heating stoves in winter, and fewer emissions during the Chinese New Year holiday in February 2019. The emissions in 2020 show a similar seasonal variation but are substantially lower than the 2019 emissions between January and March, when the 2020 emissions are 6–29% lower for $SO_2$, 8–31% lower for $NO_x$, 4–27% lower for CO, 2–26% lower for NMVOCs, 2–21% lower for $PM_{2.5}$, and 9–22% lower for BC than those emissions in 2019. The largest reductions in emissions are observed in February. The air pollutant emissions were estimated to decline by 21–31% compared to those in February 2019, with $NO_x$ illustrating the largest reductions among the air pollutants. However, after April 2020, China's emissions have gradually increased through August and finally returned to comparable levels in 2019. In August 2020, China's emissions of $SO_2$, $NO_x$, CO, NMVOCs, $PM_{2.5}$, and BC were only 6%, 1%, 2%, 1%, 2%, and 2% lower than those emissions in 2019. The sensitivity analysis of emission estimation using the same emission factors between 2019 and 2020 (values in brackets in Table 1) suggests that the decline and rebound in China's economic and industrial activities are the main drivers of the emission dynamics during the COVID-19 pandemic. With emission factors frozen, the activity growth could have driven up China's emissions more rapidly, which even surpassed the 2019 emissions by 1–4% in August 2020.

The distinct monthly variation of China's emissions in 2020 is in phase with the occurrence of both the Chinese New Year and the COVID lockdowns. The reduced socio-economic activities during the Chinese New Year holiday, which took place in February 2019 but in January 2020, explains why the emissions in January 2020 were lower than those in January 2019. However, the lower emissions in 2020 continued to persist in February and March and did not return to the normal levels until August, which is in contrast to the rapid rebound in emissions after the Chinese New Year in 2019. The larger decline and slower recovery in the monthly emissions of 2020 are coincident with the timeline of the COVID lockdowns implemented in China. The lockdown of Wuhan, where the virus outbreak was first identified, started on 23 January 2020, which set the precedent for similar control measures implemented in the other Chinese cities within the next few days. All of China's cities were placed under a strict lockdown that lasted one to two months, which substantially reduced economic activities and forced air pollutant emissions to remain much lower than the corresponding months in 2019. With the success in controlling the virus in February and March, most of China's cities gradually loosened control measures since April and Wuhan reopened on 8th April. China's economy then started to rapidly recover, reflected by the sharp rise in energy





consumption. For example, China's thermal electricity generation in April, May, and June 2020 was 1.2%, 9.0%, and 5.4%, respectively, higher than those in the same months of 2019 (Table S2). The travel demands started to resume and the road
freight transport volumes between April and August in 2020 were 1.6–6.6% higher than those in the same months of 2019.

Figures 2 and 3 present the changes in China's provincial emissions from 2019 to 2020. From Jan to April (Figure 2), all of China's provinces reduced their air pollutant emissions in 2020 compared to 2019. Hubei province, where Wuhan is located, showed the largest emission reductions due to the longest and strictest COVID lockdown measures adopted in this province. $SO_2$ and $PM_{2.5}$ emissions decreased a little more in the provinces of North and Central China, where concentrate most of the
industrial plants in China. $NO_x$ and NMVOCs emissions also presented large reductions in the provinces on China's southeast coast, which have developed and high-income economies and more vehicle ownership and transport emissions. Figure 4 illustrates the emission difference from May to August in 2020 compared to 2019, which suggests that most of the provinces except Hubei and the provinces surrounding Beijing have rapidly recovered economies and air pollutant emissions. Hubei, Beijing, and the provinces around Beijing remained their public health response systems to the COVID-19
emergency at the top level for more than three months, while the other provinces only stayed at the top level for one to two months. The stringent containment measures implemented in these provinces have hindered rebounding emissions levels.

**3.2 Drivers of emission dynamics in 2020**

The differences in the monthly emissions between 2019 and 2020 are decomposed into the contributions from power, industry, residential, and transportation sectors (Figure 4). The results suggest that the industry and transportation sectors are
the major drivers of the emissions decline between January and March in 2020. In February 2020, when China's emissions were lowest during the lockdowns, these two sectors contributed 68%, 88%, 80%, 93%, 73%, and 70%, respectively, to the emissions decrease of $SO_2$, $NO_x$, CO, NMVOCs, $PM_{2.5}$, and BC. The industry sector dominates the reductions in $SO_2$ and $PM_{2.5}$ emissions, and the industry and transportation sectors have approximately equal contributions to the decreases in $NO_x$, CO, NMVOCs, and BC emissions. The industry sector dominated emission reductions because it is the largest source of
most of these pollutants in China (Zheng et al., 2018a) and the industrial activities have dropped substantially (Table S2). The gross domestic product by industry during January and February 2020 was 13.5% lower than the corresponding months in 2019. We observe large declines in industrial products, such as cement production that dropped by 29.5% in January and February. The transportation sector revealed an even larger fall in activities, for example, road freight transport dropped by 41.5% in February 2020 than in February 2019. That explains why transport had equal or slightly higher contributions to
$NO_x$, CO, NMVOCs, and BC emission reductions than the industry sector, although it emits fewer emissions than industry.

The industry and transportation sectors were also the major drivers of emissions rebound from April to August (Figure 4). The increases in industrial emissions are coincident with the strong recovery of China's industrial economy from the COVID lockdowns. The negative growth rates of industrial productions in the first months were followed by the accelerated positive growth rates in the following months. The gross domestic product by industry has increased by 3.9–5.6% between April and





August in 2020 compared to the same months in 2019 (Table S2). Heavy industrial products such as iron, steel, cement, and aluminium have increased their monthly productions by more than 5% in August 2020 compared to August 2019. The accelerated growth of industrial activities combined with the slightly decreased emission factors made China's industrial emissions return to the emission levels of 2019. China's economic recovery also stimulated the demand for freight transport, which has substantially increased freight transport volumes on the road, rail, and inland waterways (Table S2). The transport

emissions of $NO_x$ and BC, especially those from diesel engines, have resumed the growth and reached the 2019 emission levels. However, the activities of passenger vehicles have not fully recovered yet, illustrated by the reduced passenger transport (Table S2) and the lower TOMTOM traffic index (https://www.tomtom.com/en_gb/traffic-index/, accessed on 10 November 2020). The passenger vehicles are mainly gasoline-powered and dominate CO and NMVOCs emissions in the transportation sector, therefore the transport emissions of these two pollutants were still slightly lower than those in 2019.

The residential sector is also important for the monthly variation of China's emissions in 2020, especially for $SO_2$, CO, $PM_{2.5}$, and BC. The production of service industries during January and February 2020 was 13.0% lower than the corresponding months in 2019, probably caused by the reduced commercial activities that were affected by the COVID-19 stay-at-home orders. Besides, the winter in 2020, especially February, was the second warmest one on record for the globe (https://www.climate.gov/news-features/understanding-climate/winter-and-february-2020-end-second-warmest-record-globe,

accessed on 10 November 2020). In China, the population-weighted heating degree day from January to March in 2020 was estimated to be 3.6%, 9.1%, and 3.4%, respectively, lower than the same months in 2019, suggesting that the residential heating demand was lower in 2020 compared to 2019. Therefore, the lower residential emissions between January and March in 2020 are dominated by both the reduced service industry activities during the COVID lockdowns and the lower energy demand by the residential space heating. After that, the residential emissions rebounded rapidly due to the recovered

commercial activities, revealed by the growth in the service industrial productions by 1.0–4.0% from May to August in 2020 (Table S2). The growing activities drove up air pollutant emissions in the residential sector and made residential emissions return to the 2019 emission levels.

**3.3 Comparision with observations**

Figure 5 compares the observations and emissions of $SO_2$, $NO_x$, CO, and $PM_{2.5}$ regarding the relative monthly changes from

2019 to 2020. The monthly variations of surface emissions are coincident with the pollutant concentrations observed by ground stations and TROPOMI satellite data, both of which illustrate the abrupt decline and slow rebound trends between January and August in 2020, with February showing the lowest values. The assumption of a linear decline in emission factors from 2018 to 2020 helps reduce the mismatch between observation and emission trends (comparing solid and dashed red curves in Figure 5), which suggests that the changes in activity data and emission factors both shape the emission

monthly variations in 2020. We also compare the monthly emissions with the surface observations over the three most polluted regions in China, which include the North China Plain (Figure S1), Yangtze River Delta (Figure S2), and Fenwei



plain (Figure S3). Each region has tens of cities and tens of thousands of industrial plants that emit a mass of air pollutants, which makes the anthropogenic sources dominate the air pollutant emissions from these regions. All of these three regions reveal much lower air pollutant concentrations in 2020 compared to those in 2019, and also show a nearly synchronous

decline and rebound in the pollutant concentrations from January to August, which are broadly consistent with the anthropogenic emission variations estimated in this study. Overall, the broad consistencies between emissions and observations in China and the polluted regions prove that our new emission estimation method and results can simulate the response of China's anthropogenic emissions to the COVID-19 pandemic, including the influence of the stringent lockdowns and the economic recovery after the lockdowns.

The discrepancies that still exist in the direct comparison between emissions and observations are caused by several factors. First, such a comparison is more suitable for the short-lived primary pollutants, such as $SO_2$ and $NO_x$. CO has an atmospheric lifetime of several weeks, therefore the CO abundance in the atmosphere is determined not only by the local CO emission sources but also by the regional backgrounds of CO distributions (Zheng et al., 2019), which leads to the differences in the comparisons between CO emissions and CO observations over China (Figure 5c). The ambient $PM_{2.5}$ is

composed of primary and secondary particulates, both contributing to the monthly variations in ambient $PM_{2.5}$ concentrations, which explains why the ambient $PM_{2.5}$ concentrations show a larger decline than the emissions of primary $PM_{2.5}$ during the COVID-19 lockdowns (Figure 5d). Second, the changes in meteorological conditions also remarkably influence the pollutant transport, reactions, and concentrations in the air, which is not taken into account by our comparisons in Figure 5. The meteorological impacts during the COVID lockdowns (Su et al., 2020; Zhao et al., 2020) partly contribute

to the mismatches in the direct comparisons between emissions and observations. Third, the observation data, especially satellite retrievals, are also subject to data uncertainties. The estimation of the relative changes from month to month can cancel a major part of the systematic errors in the satellite retrievals, but a substantial disturbance in the atmosphere during lockdowns including the large decrease in the aerosol concentrations could impact the data quality in the $NO_2$ column retrievals (Lin et al., 2015; Liu et al., 2020c). Last but not least, the discrepancies shown in Figure 5 also imply possible

uncertainties in our emission estimates. Uncertainties primarily lie in the emission sources without near real-time monthly statistics of activities, for example, the industrial Gross Domestic Product is used to predict the activity changes of industrial boilers and industrial solvent use, which causes uncertainties that can be reduced if more data are available in the future.

## 4 Data availability

The monthly statistics of the industrial economy and industrial products in China can be accessed from

https://data.stats.gov.cn/ (last access: 10 November 2020). The monthly statistics of the transport volume of freight and passenger by road, railways, and inland waterways in China can be downloaded from http://www.mot.gov.cn/shuju/ (last access: 10 November 2020). The monthly natural gas consumption data in China is made available at https://www.ndrc.gov.cn/fgsj/tjsj/jjyx/mdyqy/ (last access: 10 November 2020). The daily coal consumption in six major



power companies of China can be accessed from https://www.wind.com.cn/ (last access: 10 November 2020). The
TOMTOM traffic index can be downloaded from https://www.tomtom.com/en_gb/traffic-index/ (last access: 10 November
2020). The daily average temperature at 2 meter is derived from https://cds.climate.copernicus.eu/ (last access: 10 November
2020). The gridded population dataset is derived from the UN WPP-Adjusted Population Count, v4.11, which can be
accessed from https://doi.org/10.7927/H4PN93PB (last access: 10 November 2020). The reduction ratios of China's
emissions from 2019 to 2020 can be accessed from https://doi.org/10.6084/m9.figshare.c.5214920.v1 (Zheng et al., 2020).

## 5 Conclusions

We have developed a novel bottom-up approach to track China's air pollutant emissions by month through integrating the
emission model MEIC with the near real-time statistical data. We established a relation between the near real-time data with
all of the anthropogenic emission sources and the activity data used in the baseline inventory, which lay the foundation for
the extrapolation of baseline emissions to the current time. The emission estimation results based on this new approach well
simulate the abrupt decline and the slow rebound of China's air pollutant emissions from the COVID outbreak, lockdown, to
recovery in the first eight months of 2020. The observations of pollutant concentrations in the atmosphere independently
evaluate the estimates of monthly emissions variation, while the discrepancies illustrated in the direct comparisons also
imply possible uncertainties that deserve more attention in the future. The dynamic emission inventories built using such
methods can provide timely emissions input to chemical transport models, which mitigates the map between the real-time
observations and the time-lagged emissions data during COVID-19 (Huang et al., 2020). The method developed here can be
applied to other regions in the world, where the near real-time data are available and can be linked to the baseline bottom-up
emission inventory systematically. This is a unique opportunity to understand the dynamic energy-emission-air quality
cascade in a short time and probably also to imply effective mitigation strategies in the future. Perhaps more importantly, the
methods developed here, as well as in our previous paper (Zheng et al., 2020), constitutes a pragmatic effort to monitor
emissions in near real-time, not just for the analysis of the ongoing COVID-19 pandemic but also for enhancing our abilities
in the future to track emission mitigation progress toward air quality goals in a rapidly changing economic situation.

## Supplement

The supplement related to this article has two tables (i.e., Table S1 and Table S2), which are available online.

## Author contributions

BZ and QZ designed the study. BZ developed the new emission estimation method and estimated China's anthropogenic
emissions. The manuscript was written by BZ and QZ and was revised and discussed by all of the coauthors.



**Competing interests.**

The authors declare that they have no conflict of interest.

**Acknowledgements.**

280   This work was supported by the National Natural Science Foundation of China (41921005 and 41625020).





**Table 1. China's monthly anthropogenic emissions between January and August in 2019 and 2020.** The values in brackets represent the emission estimation results in 2020 using the same emission factors as in 2019.

| | SO$_2$ | NO$_x$ | CO | NMVOCs | PM$_{2.5}$ | BC |
|---|---|---|---|---|---|---|
| **Monthly emissions in 2019 (Tg)** | | | | | | |
| January | 0.84 | 1.77 | 14.33 | 2.33 | 0.71 | 0.13 |
| February | 0.65 | 1.44 | 11.32 | 2.03 | 0.56 | 0.10 |
| March | 0.71 | 1.76 | 11.31 | 2.35 | 0.55 | 0.10 |
| April | 0.55 | 1.62 | 8.49 | 2.07 | 0.42 | 0.07 |
| May | 0.54 | 1.63 | 8.33 | 2.10 | 0.41 | 0.07 |
| June | 0.57 | 1.73 | 8.57 | 2.33 | 0.42 | 0.07 |
| July | 0.55 | 1.66 | 8.28 | 2.07 | 0.41 | 0.07 |
| August | 0.56 | 1.68 | 8.36 | 2.08 | 0.41 | 0.07 |
| Total | 4.97 | 13.29 | 78.99 | 17.36 | 3.89 | 0.68 |
| **Monthly emissions in 2020 (Tg)** | | | | | | |
| January | 0.79 (0.85) | 1.63 (1.69) | 13.70 (13.85) | 2.28 (2.29) | 0.70 (0.71) | 0.12 (0.13) |
| February | 0.46 (0.50) | 0.98 (1.02) | 8.22 (8.30) | 1.50 (1.51) | 0.44 (0.45) | 0.08 (0.08) |
| March | 0.62 (0.68) | 1.55 (1.61) | 9.90 (10.03) | 2.13 (2.14) | 0.51 (0.53) | 0.09 (0.09) |
| April | 0.50 (0.56) | 1.56 (1.62) | 7.82 (7.95) | 1.95 (1.96) | 0.40 (0.42) | 0.07 (0.07) |
| May | 0.50 (0.56) | 1.60 (1.66) | 7.94 (8.08) | 2.02 (2.04) | 0.40 (0.42) | 0.06 (0.07) |
| June | 0.53 (0.59) | 1.72 (1.79) | 8.49 (8.62) | 2.31 (2.32) | 0.41 (0.43) | 0.07 (0.07) |
| July | 0.51 (0.56) | 1.62 (1.68) | 8.07 (8.21) | 2.03 (2.05) | 0.40 (0.42) | 0.06 (0.07) |
| August | 0.52 (0.58) | 1.67 (1.74) | 8.22 (8.36) | 2.05 (2.06) | 0.40 (0.42) | 0.07 (0.07) |
| Total | 4.43 (4.88) | 12.33 (12.81) | 72.36 (73.40) | 16.27 (16.37) | 3.66 (3.80) | 0.62 (0.65) |
| **Change in emissions from 2019 to 2020** | | | | | | |
| January | −6% (1%) | −8% (−4%) | −4% (−3%) | −2% (−1%) | −2% (0%) | −9% (−6%) |
| February | −29% (−23%) | −31% (−29%) | −27% (−27%) | −26% (−26%) | −21% (−20%) | −22% (−20%) |
| March | −13% (−5%) | −12% (−8%) | −12% (−11%) | −9% (−9%) | −8% (−5%) | −10% (−7%) |
| April | −9% (2%) | −4% (0%) | −8% (−6%) | −6% (−5%) | −5% (0%) | −5% (−2%) |
| May | −7% (4%) | −2% (2%) | −5% (−3%) | −3% (−3%) | −3% (2%) | −4% (0%) |
| June | −7% (4%) | 0% (3%) | −1% (1%) | −1% (0%) | −2% (3%) | −2% (2%) |
| July | −8% (2%) | −2% (1%) | −3% (−1%) | −2% (−1%) | −3% (2%) | −2% (1%) |
| August | −6% (4%) | −1% (3%) | −2% (0%) | −1% (−1%) | −2% (3%) | −2% (1%) |
| Total | −11% (−2%) | −7% (−4%) | −8% (−7%) | −6% (−6%) | −6% (−2%) | −8% (−5%) |



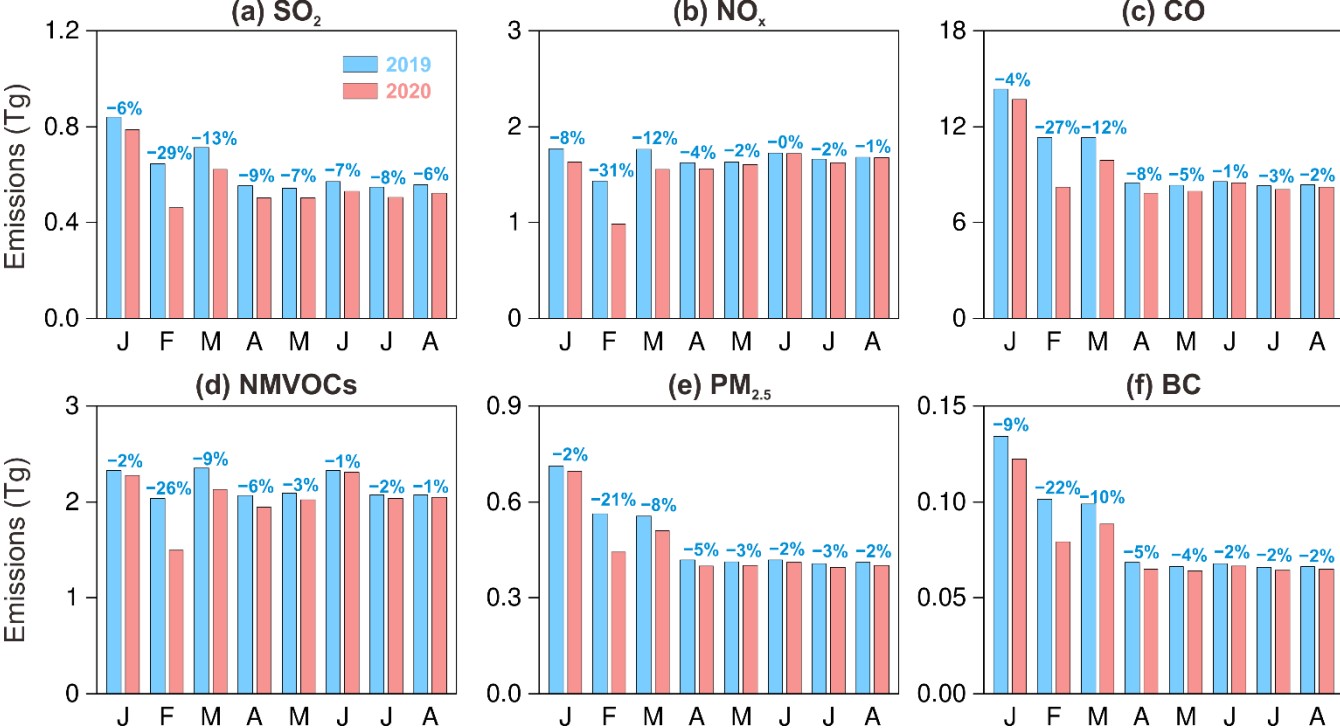

**Figure 1: Monthly emissions from January to August in 2019 and 2020.** Six pollutants are presented including $SO_2$ (a), $NO_x$ (b), CO (c), NMVOCs (d), $PM_{2.5}$ (e), and BC (f). The blue bars represent the emissions in 2019, and the red bars represent the emissions in 2020. The values above the bars represent the change in monthly emissions in 2020 compared to the same month in 2019.



**Figure 2: Differences in January–April emissions between 2019 and 2020.** Four pollutants are presented including SO$_2$ (a), NO$_x$ (b), NMVOCs (c), and PM$_{2.5}$ (d). The background map shows the provincial borders in China. The colour of each province represents the emissions from January to April in 2020 minus the emissions from January to April in 2019. The grey colour represents the province whose data are not estimated in this study.

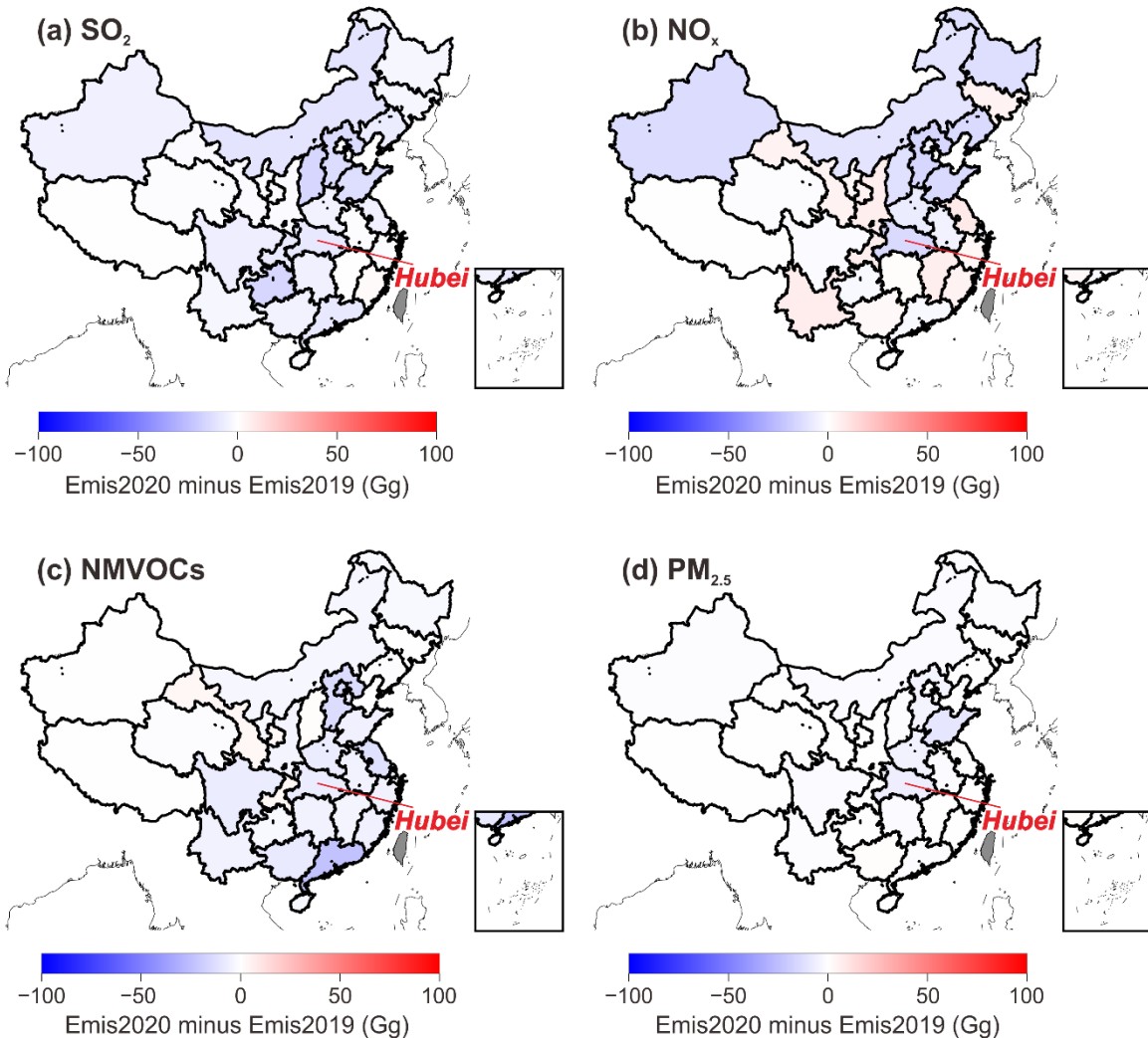

**Figure 3: Differences in May–August emissions between 2019 and 2020.** Four pollutants are presented including $SO_2$ (a), $NO_x$ (b),
NMVOCs (c), and $PM_{2.5}$ (d). The background map shows the provincial borders in China. The colour of each province represents the
emissions from May to August in 2020 minus the emissions from May to August in 2019. The grey colour represents the province whose
data are not estimated in this study.



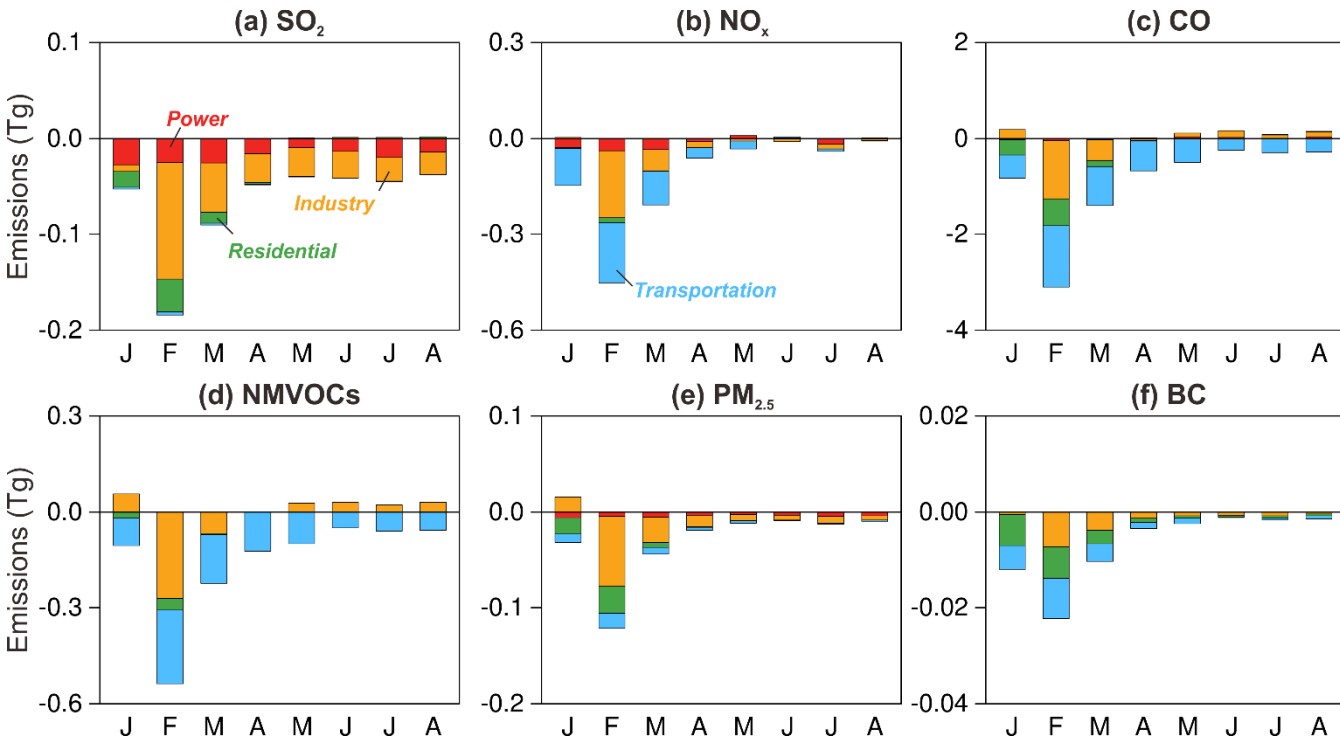

**Figure 4: Differences in monthly emissions between 2019 and 2020 by source sector.** Six pollutants are presented including $SO_2$ (a),
$NO_x$ (b), CO (c), NMVOCs (d), $PM_{2.5}$ (e), and BC (f). Each bar represents the results of monthly emissions in 2020 minus those in 2019.
The color of stacked bars represents the source sectors of power (red), industry (yellow), residential (green), and transportation (blue).



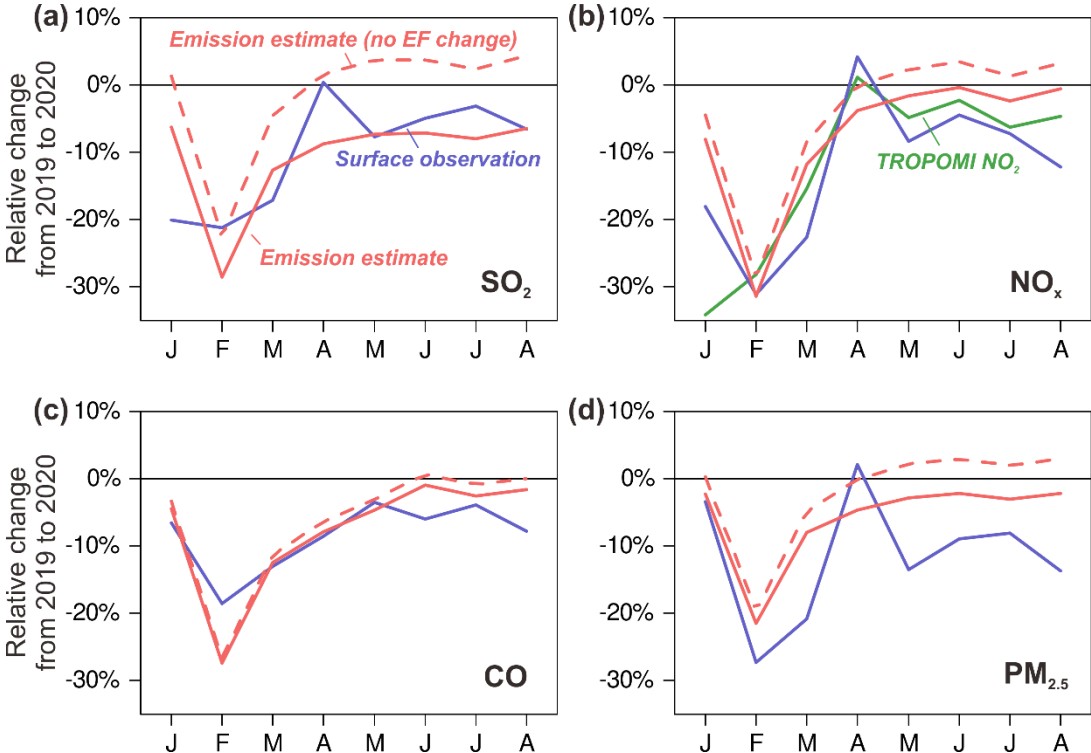

**Figure 5: Comparison between observations and emissions regarding the monthly relative changes from 2019 to 2020.** Four pollutants are presented including $SO_2$ (a), $NO_x$ (b), CO (c), and $PM_{2.5}$ (d). The red solid curves represent the emission results of this study; the red dashed curves represent sensitivity estimation of emissions using the same emission factors between 2019 and 2020; the purple curves represent the surface observations (http://106.37.208.233:20035/, accessed on 10 November 2020); the green curve in (b) represents the TROPOMI $NO_2$ column retrievals (http://www.temis.nl/airpollution/no2col/data/tropomi/, accessed on 10 November 2020).

305



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
