# Peer review of "Changes in China's anthropogenic emissions and air quality during the COVID-19 pandemic in 2020"

_Earth System Science Data, 2020_

## Referee Comment (RC1) · Anonymous Referee #1 · 7 Jan 2021

The authors develop a simple method based on the most recent statistical data for estimating the anthropogenic emissions of air pollutants in China during the period from January to August in 2020. They report for the first time the changes in air pollutants emissions caused by the COVID-19 lockdowns in China using a bottom-up approach. Additionally, the relative changes in monthly emissions from 2019 to 2020 are compared with the satellite and ground-based observations. The emission datasets developed in this study provide essential and important information for the analysis of the COVID-19 pandemic in China.

Consequently, the contents of this manuscript and datasets developed in this study are suitable for "Earth System Science Data". However, there are some points which should be analyzed and clarified. The reviewer recommends the acceptance of this

manuscript after minor revisions.

(Major comments) (1) Lines 100-112: To what extent can the simple method developed in this study reproduces the changes of emissions in the past years? For example, by comparing with the MEIC in the emission changes from 2018 to 2019, it might be possible to validate the method and estimate its uncertainties. Such analysis should be added for identifying the application of the method to other cases and other regions.

(2) Figure 4: Some differences among six pollutants are found in the industrial emissions. The values for CO and NMVOCs are small positive in January and May to August while the values for $SO_2$, $NO_x$, and BC are negative in the same period and the value for $PM_{2.5}$ is positive in January only. These differences should be discussed.

(3) Figure 5: In January, there are big differences between emissions and observations for $SO_2$ and $NO_x$ while their differences for CO and $PM_{2.5}$ are smaller. The authors should discuss the reasons more carefully. The regional background largely affected the observed CO as the authors pointed out in lines 227-229. However, it is surprising that the differences between emissions and observations are relatively small in Figure 5(c). Further discussion is needed.

(Minor comments) (1) Lines 118-120: Is there no observation data of NMVOC or NMHC concentration in China?

(2) Line 128: For NMVOC (and $NO_x$), the emission declining from January to August seems to be not found in Figure 1.

(3) Line 210: Is "surface emissions" correct?

(4) Lines 226-229: It looks like small differences between emissions and observations in Figure 5d. If the effects of regional background are large, the differences may be more increasing.

(5) Figure 1: It's better that the monthly emissions are decomposed into source sectors like Figure 4.

---

## Referee Comment (RC2) · Anonymous Referee #2 · 7 Jan 2021

Zheng et al. (2020) developed a bottom-up approach to estimate anthropogenic emissions over mainland China during and after covid-19 lockdown. The results suggest the reduced anthropogenic emissions due to covid-19 lockdown are mainly from industry and transportation sectors. Despite all the merits of this approach mentioned in the manuscript, the emission estimates need thoroughly evaluated to better support the conclusion. Therefore, the reviewer recommends a major revision before accepted for publication.

General comments:

In this work, changes in the emission are evaluated against changes in surface observations and satellite retrievals. However, the changes in the surface concentrations do not necessarily reflect the similar changes in the emission. There are many processes/factors that could affect surface concentrations. This kind of evaluation does not provide much information on the uncertainty of the emission estimates. As mentioned in the manuscript, meteorology plays a significant role on surface concentrations, which is not considered in this work. A better way to evaluate the emission estimates would be comparing surface concentrations from an emission-driven model simulation with surface observations. It would be interesting to see a combination of a top-down approach (via observational constraints) and a bottom-up approach (used in this work) to better assess the emission estimates and to add more value to this work.

Specific comments:

Page 3, line 75-76, are the emissions from cooking included in residential sector?

Page 3, line 89-94, does EF2019/EF2018 have monthly variability? Or should EFm2019/EFm2018 be used?

Page 4, line 106-108, could you explain "assumption of no change" to "predict the 2019-to-2020 change"? Just curious, do you have estimates in cooking sources? Should be higher in 2020 than 2019?

Page 5, line 141-142, is it possible to separate the impacts from Chinese New Year and COVID lockdown? As mentioned in the manuscript, one happened in Feb 2019 and one in Jan 2020. It would be interesting to see the impacts from COVID lockdown only.

Page 6, line 178, do you have estimates for aviation emissions?

Figure 4, any explanations on higher industrial sources for CO, NMVOCs, and PM2.5, in Jan 2020 than Jan 2019?

Figure 5, see general comments.

---

## Author Comment (AC1) · 11 Apr 2021

**Referee #1**

The authors develop a simple method based on the most recent statistical data for estimating the anthropogenic emissions of air pollutants in China during the period from January to August in 2020. They report for the first time the changes in air pollutants emissions caused by the COVID-19 lockdowns in China using a bottom-up approach. Additionally, the relative changes in monthly emissions from 2019 to 2020 are compared with the satellite and ground-based observations. The emission datasets developed in this study provide essential and important information for the analysis of the COVID-19 pandemic in China.

Consequently, the contents of this manuscript and datasets developed in this study are suitable for "Earth System Science Data". However, there are some points which should be analyzed and clarified. The reviewer recommends the acceptance of this manuscript after minor revisions.

**Response:**

We appreciate the referee's positive and constructive comments. Our point-by-point responses are given as follows.

(Major comments) (1) Lines 100-112: To what extent can the simple method developed in this study reproduces the changes of emissions in the past years? For example, by comparing with the MEIC in the emission changes from 2018 to 2019, it might be possible to validate the method and estimate its uncertainties. Such analysis should be added for identifying the application of the method to other cases and other regions.

**Response:**

We thank the reviewer's suggestions and agree that the validation of our method will identify the application possibilities in other cases. Our method relied on the monthly statistical data to track emissions, which have already been used by the MEIC model to reconstruct the monthly variation of emissions in the past years. The only difference is that MEIC has more constraints on energy consumption and emission factors in the past years than in the COVID period. To evaluate our method and the uncertainties based on an independent method and dataset, we discussed with the editor and decided to run an air quality model driven by our estimated emissions to simulate the interannual changes in air pollutant concentrations and evaluate the simulation results against surface observations. The comparison results reveal a broad consistency, suggesting

that our emission estimates can reproduce air pollution changes well. The discrepancies and implications for uncertainties have also been discussed in the revised manuscript.

(2) Figure 4: Some differences among six pollutants are found in the industrial emissions. The values for CO and NMVOCs are small positive in January and May to August while the values for $SO_2$, $NO_x$, and BC are negative in the same period and the value for $PM_{2.5}$ is positive in January only. These differences should be discussed.

**Response:**

The emissions of air pollutants tended to be lower in 2020 than in 2019 due to the reduced industrial activities during the lockdown. However, the activities of part of the industrial sources before and after the lockdown were larger in 2020 than in 2019, which drove up emissions of specific air pollutants. For example, the productions of iron, steel, and non-ferrous metals during January and February were 3.1%, 3.1%, and 2.2%, respectively, higher in 2020 than those in 2019, which have generated higher emissions of CO and $PM_{2.5}$ in January 2020. The productions of iron and steel from May to August in 2020 were 2.4–9.1% higher than the corresponding months in 2019, leading to higher CO emissions in 2020. The productions of crude oil and petrochemical products such as ethylene during January and February were 3.7% and 5.6% higher in 2020 than those in 2019, which explains the higher NMVOCs emissions in Jan 2020. From May to August, the productions of crude oil and the total volume of crude oil refineries process were 0.6–12.4% higher in 2020 than in 2019, which caused more NMVOCs emissions. These monthly changes in industrial activities have been shown in Table S2, which have also been clarified in the main text of the revised manuscript.

(3) Figure 5: In January, there are big differences between emissions and observations for $SO_2$ and $NO_x$ while their differences for CO and $PM_{2.5}$ are smaller. The authors should discuss the reasons more carefully. The regional background largely affected the observed CO as the authors pointed out in lines 227-229. However, it is surprising that the differences between emissions and observations are relatively small in Figure 5(c). Further discussion is needed.

**Response:**

To account for the impact of regional background, we run the air quality model WRF-CMAQ to simulate surface concentrations of air pollutants and compared the changes in modeled concentrations to surface observations in Figure 5 in the revised manuscript.

The comparison suggests that the model simulations driven by our estimated emissions reproduced the changes in surface observations well. The results also reveal some discrepancies between simulations and observations, probably caused by uncertainties in emissions and modeling, which have been discussed in the revised manuscript.

(Minor comments) (1) Lines 118-120: Is there no observation data of NMVOC or NMHC concentration in China?

**Response:**

Not yet, the surface measurement network in China does not report NMVOC or NMHC.

(2) Line 128: For NMVOC (and $NO_x$), the emission declining from January to August seems to be not found in Figure 1.

**Response:**

This sentence has been rewritten as follows.

"China's emissions of $SO_2$, CO, $PM_{2.5}$, and BC in 2019 reveal an evident seasonal variation with emissions declining from January to August…"

(3) Line 210: Is "surface emissions" correct?

**Response:**

We have changed "surface emissions" to "anthropogenic emissions" in the revised manuscript.

(4) Lines 226-229: It looks like small differences between emissions and observations in Figure 5d. If the effects of regional background are large, the differences may be more increasing.

**Response:**

Please refer to our response to the major comment (3).

(5) Figure 1: It's better that the monthly emissions are decomposed into source sectors like Figure 4.

**Response:**

Done.

---

## Author Comment (AC2) · 11 Apr 2021

**Referee #2**

Zheng et al. (2020) developed a bottom-up approach to estimate anthropogenic emissions over mainland China during and after covid-19 lockdown. The results suggest the reduced anthropogenic emissions due to covid-19 lockdown are mainly from industry and transportation sectors. Despite all the merits of this approach mentioned in the manuscript, the emission estimates need thoroughly evaluated to better support the conclusion. Therefore, the reviewer recommends a major revision before accepted for publication.

**Response:**

We thank the referee for his/her effort to improve our manuscript. We provide point-by-point responses to the comments as follows.

General comments:

In this work, changes in the emission are evaluated against changes in surface observations and satellite retrievals. However, the changes in the surface concentrations do not necessarily reflect the similar changes in the emission. There are many processes/factors that could affect surface concentrations. This kind of evaluation does not provide much information on the uncertainty of the emission estimates. As mentioned in the manuscript, meteorology plays a significant role on surface concentrations, which is not considered in this work. A better way to evaluate the emission estimates would be comparing surface concentrations from an emission-driven model simulation with surface observations. It would be interesting to see a combination of a top-down approach (via observational constraints) and a bottom-up approach (used in this work) to better assess the emission estimates and to add more value to this work.

**Response:**

We agree with the reviewer that it is a better way to evaluate our emission estimates through a model simulation. In the revised manuscript, we have run the air quality model WRF-CMAQ driven by our estimated emissions for 2019 and 2020, simulated the interannual changes in surface concentrations of air pollutants, and compared the modeling results with surface observations in Fig. 5. The comparison results suggest that the model simulations driven by our estimated emissions reproduced the changes in surface observations well. The text in the discussions has been revised accordingly.

Following the reviewer's suggestions integrating top-down and bottom-up approaches, we have collected the top-down estimated air pollutant emissions from previous literature, which are broadly consistent with our bottom-up estimates. The consistency between top-down and bottom-up results proves the reliability of the emission results.

Specific comments:

Page 3, line 75-76, are the emissions from cooking included in residential sector?

**Response:**

Yes, the emissions from cooking stoves have been included in the residential sector.

Page 3, line 89-94, does EF2019/EF2018 have monthly variability? Or should EFm2019/EFm2018 be used?

**Response:**

Only the power sector in the MEIC model has monthly variability in EF2019/EF2018, mainly caused by the improvement of air pollution control efficiencies (e.g., via installing new devices). Since the change in emission factors may not occur in the same month of different years, we use the ratio of annual average emission factors to represent the continuous improvement in the air pollution control of power plants. For the other emission sources, we do not have monthly variability in the EF2019/EF2018.

Page 4, line 106-108, could you explain "assumption of no change" to "predict the 2019-to-2020 change"? Just curious, do you have estimates in cooking sources? Should be higher in 2020 than 2019?

**Response:**

Yes, we have estimated the emissions from cooking sources, the activities of which have been estimated based on population and energy consumed for cooking per person. The food demand per person and the associated energy use for cooking are relatively constant for two consecutive years. Therefore, we assume that both of these two factors remain unchanged in 2019 and 2020, resulting in an assumption of no change to predict the 2019-to-2020 change in cooking activities and emissions from cooking sources.

Page 5, line 141-142, is it possible to separate the impacts from Chinese New Year and COVID lockdown? As mentioned in the manuscript, one happened in Feb 2019 and one in Jan 2020. It would be interesting to see the impacts from COVID lockdown only.

**Response:**

The COVID lockdown overlapped with the Chinese New Year in 2020. For example, the lockdown measures in Wuhan started two days before the Chinese New Year and lasted for two months covering the entire holiday of Chinese New Year, making it difficult to separate the impacts from the holiday and COVID lockdown on emissions.

Page 6, line 178, do you have estimates for aviation emissions?

**Response:**

No, the aviation emissions are not included in our estimates.

Figure 4, any explanations on higher industrial sources for CO, NMVOCs, and $PM_{2.5}$, in Jan 2020 than Jan 2019?

**Response:**

The higher industrial source emissions of CO, NMVOCs, and $PM_{2.5}$ are due to the larger industrial activities in Jan 2020 (especially before the lockdown) than in Jan 2019. The productions of iron, steel, and non-ferrous metals during January and February are 3.1%, 3.1%, and 2.2% higher in 2020 than those in 2019, which are the major driver of higher emissions of CO and $PM_{2.5}$. The productions of crude oil and petrochemical products such as ethylene during January and February are 3.7% and 5.6% higher in 2020 than those in 2019, which explains the higher NMVOCs emissions in Jan 2020.

Figure 5, see general comments.

**Response:**

Please refer to our responses to the general comments.

---

## Referee Report (RR1)

The authors have addressed all the comments and the manuscripts has been improved after the revision. The reviewer recommends acceptance after minor revision.

General comment:
A new emission tool has been developed in this work, which generates emissions over Mainland China at a monthly basis and at provincial level. Is it possible to increase the temporal and spatial resolution in this emission tool? In other words, is it possible to apply your emission estimates to study urban scale air quality or air pollution events? In that sense, we may need separate estimates for weekdays/weekends and urban/rural areas. As for the covid-19 impacts, it would also be interesting to see how different the impacts are over urban and rural areas. Also, if we want to use those emissions to drive the chemical transport model, how do you deal with the spatial resolution issue?

Specific comments:
Section 2.3, what meteorological input is used for the model? I assume they are different for 2019 and 2020, is that correct?

Figure 6, there is an overall agreement on the relative changes of air pollutants between model estimates and surface observations. One thing we need pay attention is that the differences in meteorology between 2019 and 2020 could also play a role here. Also, it looks like the correlation is poorer over Yangtze River Delta than North China Plain. Is that something related to much larger uncertainties in the emissions over Yangtze River Delta than North China Plain or more likely due to inherent model uncertainties?

Page 9, line 255-261, any explanations on such a large difference in NOx emission reductions between your estimates and Zhang et al. (2021)?

---

## Author Response (AR2)

The authors have addressed all the comments and the manuscripts has been improved after the revision. The reviewer recommends acceptance after minor revision.

**Response:**

We thank the referee's positive comments on our revised manuscript. Our point-by-point responses are given below.

General comment:

A new emission tool has been developed in this work, which generates emissions over Mainland China at a monthly basis and at provincial level. Is it possible to increase the temporal and spatial resolution in this emission tool? In other words, is it possible to apply your emission estimates to study urban scale air quality or air pollution events? In that sense, we may need separate estimates for weekdays/weekends and urban/rural areas. As for the covid-19 impacts, it would also be interesting to see how different the impacts are over urban and rural areas. Also, if we want to use those emissions to drive the chemical transport model, how do you deal with the spatial resolution issue?

**Response:**

This work generated emissions by month and by province because the near real-time data we can access is only available at monthly and provincial levels at present. It is possible to increase the temporal and spatial resolutions in our emission tool once we get the data of higher resolutions. To support chemical transport model simulations, we allocated our estimated emissions in 2020 to higher resolutions based on the source-specific emission maps of the most recent year, which were derived from the MEIC 2019 emissions in this study.

Specific comments:

Section 2.3, what meteorological input is used for the model? I assume they are different for 2019 and 2020, is that correct?

**Response:**

The meteorological fields were simulated by the WRF model, which was initialized by the NCEP FNL Operational Model Global Tropospheric Analyses dataset. The meteorological fields were simulated for 2019 and 2020 separately. We have clarified it in the revised manuscript.

Figure 6, there is an overall agreement on the relative changes of air pollutants between model estimates and surface observations. One thing we need pay attention is that the differences in meteorology between 2019 and 2020 could also play a role here. Also, it looks like the correlation is poorer over Yangtze River Delta than North China Plain. Is that something related to much larger uncertainties in the emissions over Yangtze River Delta than North China Plain or more likely due to inherent model uncertainties?

**Response:**

The poorer model performance over Yangtze River Delta than North China Plain is possibly caused by the inherent uncertainties in the simulation of meteorological fields, which involves larger uncertainties over a coastal and complex terrain area (e.g., Yangtze River Delta) than in a plain area (e.g., North China Plain).

Page 9, line 255-261, any explanations on such a large difference in NOx emission reductions between your estimates and Zhang et al. (2021)?

**Response:**

We have compared our bottom-up estimates of $NO_x$ emission reductions with different top-down inversions in the main text. Only the comparison with Zhang et al. (2021) shows a large difference. They estimated a decline of 53.4% in China's $NO_x$ emissions during COVID lockdown than the same period in 2019, which is much larger than the observed decrease of about 30% in the surface $NO_2$ concentrations over China. It is difficult to identify the primary source of discrepancy shown in these comparisons because it could be caused by uncertainties in emission inventory parameters, inverse modeling framework, and the satellite retrievals used to constrain the inverse models.